# Maternal employment and child nutritional status in Uganda

**Olivia Nankinga**[1]*, **Betty Kwagala**[1], **Eddy J. Walakira**[2]

**1** Department of Population Studies, School of Statistics and Planning, Makerere University, Kampala, Uganda, **2** Department of Social Work and Social Administration, Makerere University, Kampala, Uganda

☉ These authors contributed equally to this work.
* onankinga@gmail.com

**Data Availability Statement:** All Uganda Demographic and Health Survey (UDHS) - UGKR data files are available at the DHS program website (https://www.dhsprogram.com/data/dataset/Uganda_Standard-DHS_2016.cfm?flag=0) upon registration.

## Abstract

Nearly half of all deaths among children under five (U5) years in low- and middle-income countries are a result of under nutrition. This study examined the relationship between maternal employment and nutrition status of U5 children in Uganda using the 2016 Uganda Demographic and Health Survey (UDHS) data. We used a weighted sample of 3531 children U5 years born to working women age 15–49. Chi-squared tests and multivariate logistic regressions were used to examine the relationship between maternal employment and nutritional outcomes while adjusting for other explanatory factors. Results show that children whose mothers had secondary education had lower odds of stunting and underweight compared with children whose mothers had no formal education. Children who had normal birth weight had lower odds of stunting, wasting and being underweight compared with children with low birth weight. Children whose mothers engaged in agriculture and manual work had higher odds of stunting compared with those whose mothers engaged in professional work. Additionally, children whose mothers were employed by nonfamily members had higher odds of wasting and being underweight compared with children whose mothers were employed by family members. Other determinants of child nutritional status included region, age of the mother, and age and sex of the child. Interventions aimed at improving the nutritional status of children of employed women should promote breastfeeding and flexible conditions in workplaces, target those of low socio-economic status and promote feeding programs and mosquito net use for both mothers and children.

## Introduction

Child nutrition outcomes are important in measuring the socio-economic development of a country [1]. Nutrition status contributes to child health outcomes [2]. This is emphasized in the sustainable development goal (SDG) 2 and 3 which aim at ending hunger, achieving food security, improving nutrition and ensuring healthy lives and wellbeing for all the population.

Global mortality estimates show that nearly half of all deaths among children under age 5 result from malnutrition [3, 4]. Most of such deaths occur in Africa and Asia. In 2018, 22% of U5 children globally were stunted, 7% were wasted and 3% were severely wasted. Likewise,

**Funding:** ON received DAAD PhD Scholarship (Reference Number: 91602002) from the Germany Academic Exchange Service, https://www.daad.or. ke/en/about-us/daad-regional-office-nairobi/daad-in-uganda/ and Next Generation Social Sciences in Africa: Doctoral Dissertation Proposal Fellowship 2015 from the Social Science Research Council, https://www.ssrc.org. The funders had no role in study design, data collection and analysis, decision to publish, or preparation of the manuscript.

**Competing interests:** The authors have declared that no competing interests exist.

over 90% of these stunted and wasted children live in Africa and Asia [5, 6]. In sub-Saharan Africa, the prevalence of stunting and wasting is 33% and 8% in the same period respectively. Sub-Saharan Africa is the only region that registered an increase in stunting; 16% of U5 children were underweight [7, 8].

The 2016 Uganda Demographic and Health Survey (UDHS) showed that, 29% of the children were stunted, 4% were wasted and 11% were underweight. Between 2011 and 2016, the prevalence of anaemia increased from 49% to 53%. Close to four in ten of the children (38%) were vitamin A deficient and only 6% of the children were fed appropriately as recommended by the infant and young child feeding guidelines [9, 10].

Undernutrition increases the frequency and severity of infections, delays recovery and increases the risk of death from such infections. In early childhood, under nutrition can lead to stunting, poor development and cognitive outcomes, poor academic performance and reduced work productivity in adulthood [11]. Apart from nutritional intake other factors associated with the child nutritional status include the characteristics of the community, household, mother and child [12–14].

Poverty has adverse effects on nutrition of household members. Households in the poorest wealth quintiles have the highest prevalence of child undernutrition [15–17]. In addition, poverty affects access to and utilisation of health care services [13]. This increases the risk of child morbidity and mortality.

Weight at birth (which is largely a result of maternal nutrition), gender and recent illness episodes are among the key predictors of child health outcomes [16, 18–20]. Birth weight, illnesses, accessibility and utilisation of health services and feeding are associated with child health outcomes [21]. Gender differences are attributed to gender preferences in food allocation [22].

Maternal factors such as age have been associated with child health outcomes [18, 23]. Older mothers have been observed to have healthier children than younger mothers. High maternal education attainment has a mitigating effect on child health outcomes; with reduced odds of child stunting, wasting and being underweight of mothers with higher levels of education compared with uneducated mothers [13, 24].

Globally, the proportion of women in employment declined from 50 percent in 2014 [25] to 49 percent in 2018 [26]. Even with the decline, the gender gap in employment is gradually closing. In developing countries including sub-Saharan African countries, the proportion of women in employment at 69% was higher than the global rate at 65%. Some studies have shown that maternal employment reduces time spent on child care owing to the demands of employment [27]. However, other studies have shown that the time spent at work does not necessarily reduce the time spent in physical and interactive care for the child [28]. Earnings from employment also enhance child nutrition and health [29].

Maternal employment empowers women economically and socially and is in line with the sustainable development goal 8 which aims at promoting economic growth and productive employment for all. Additionally, sustainable development goal 2 aims at (among others) ending hunger, achieving food security, and improving nutrition [30] which can significantly be addressed through maternal employment. To achieve this, one of Uganda's National Development Plan II key targets aims at ensuring that people have access to healthier and sustainable diets so as to eradicate all forms of malnutrition, in an effort to propel the country towards middle income status [31].

Maternal employment has a bearing on both child and maternal health. Some of the principal pathways through which maternal employment affects child nutrition include level of income and child care practices [32]. Income generated and controlled by a women plays a key role in contributing to child and household food as well as health budget [33]. On the

hand, employment that entails absence of the mother usually implies partial weaning or cessation of breast feeding and inability to monitor child feeding and care [34].

Despite the presence of international and local frameworks on nutrition, malnutrition remains a big challenge in Uganda. This paper examined the association between maternal employment and nutritional outcomes of U5 children in Uganda.

## Data and methods

### Data source

The study used secondary data from the 2016 Uganda Demographic and Health Survey (UDHS). The UDHS is a nationally representative survey of the country. The survey interviewed 18506 women from all regions in the country. Details of how the sample was drawn may be found in the UDHS report [10]. Weight and height of U5 children was measured in a subsample of one-third of households. For this study, we used the children's recode (UGKR file) which contains data on U5 children (0–59 months). The dataset contains information on pregnancy, postnatal care, immunization and health of the child. It further contains data on the background characteristics, work and reproductive health of the mother among other factors. Only U5 children born to employed women whose height and weight were measured during the survey were considered for the analysis. In the UDHS, employed women are women who worked seven days preceding the survey and those who did not work in the seven days but are regularly employed and were absent from work for leave, illness, vacation or any related reason. Consideration of these criteria resulted into 3531 weighted cases.

### Variables and their measurements

**Outcome variable.** For this study, child nutritional status was measured using the three indices of anthropometric indicators. These indices are computed from anthropometric data and include height for age, weight for height and weight for age. All children whose height for age, weight for height or weight for age were below minus two standard deviations (-2 SD) from the median of the reference population were classified as stunted, wasted and underweight respectively. All children whose result was -2 SD or above were classified as "Not" for each of the three outcomes [10, 12, 15].

Stunting is a chronic sign of undernutrition that reflects inadequate nutrition, recurrent infection, and inadequate psychosocial stimulation. Wasting results from inadequate food intake or recent illnesses that cause weight loss and consequently acute undernutrition. Underweight is a measure of both acute and chronic nutrition.

**Explanatory variables.** Maternal employment was categorized as 1 "Professional or formal" 2 "sales and services" 3 "Agriculture and Manual Work" and 4 "Domestic or household work". Category 1 included professional, managerial, technical and clerical support work. Category 2 included sales and services. Category 3 included food growing, forestry, fishery and related labourers. Category 4 included plant and machinery operators, drivers, cleaners, labourers in mining and construction and other manual work. Category 5 included household work for example caregiving work. Maternal employer was captured from the question of who the respondent worked for. Responses were categorised as: a family member, someone else—nonrelative or were self-employed. Household decision making with respect to woman's own health was considered in this study. All women who reported that they individually or jointly with their spouses participated in decision making were recoded as 1 "involved in decision making". All cases where the respondent did not participate in decision making concerning her health were recoded into 0 "not involved". Respondents were asked whether distance to health facility was a challenge to them. These were recoded as 0 for "not a big problem" and 1

for "a big problem". Maternal age was recoded into three categories: 15–24, 25–34 and 35–49. Region was recoded into four regions namely central, east, north and west.

Age of the child was recoded as "0–23" for children who were aged between 0 and 23 months and "24–59" for children aged between 24 and 59 months.

Deworming was captured from the question about whether the child had been given drugs for intestinal worms during the 6 months preceding the survey. All children who were reported to have been given drugs were recoded as 1 "Yes" and all others were recoded into 0 "No". The "No" category included children who had not been given the drugs, whose mothers did not know whether the children were given the drugs and the missing cases. Vitamin A supplementation in the six months preceding the survey was recoded as 1 "Yes", for all children who received Vitamin A, and 0 "No" for those that did not. Concerning sickness that occurred two weeks to the survey, all children who suffered from any of the three sicknesses (diarrhoea, fever or cough) were recoded as 1 "Yes" and those who had none of the three were recoded as 0 "No".

The type of toilet facility was categorized as improved, shared and unimproved toilet facilities. Improved facilities only included flush toilets connected to sewer systems, septic tanks and pit latrines, ventilated improved pit latrines, pit latrines with slabs and composting toilets that were not shared. Shared facilities included the all toilets of the above categories that are used by more than one household. Unimproved facilities included all pit latrines without slabs, hanging and bucket toilets and those without toilet facilities. The source of drinking water was recoded into 1 "improved sources" and 0 "unimproved sources". The number of U5 children in the household was recoded as 1, 2 and 3 for one child, two children, and three or more children respectively. Wealth status, educational attainment, marital status, employer and sex of the child variables were used as coded in the UDHS datasets.

## Statistical analysis

Data were weighted using the women's individual sample weight (v005) to cater for nonresponse and disproportionate sample selection. The stata survey command "svy set" was used to cater for the complex survey design that is applied in DHS data collection. Data were analysed at the univariate, bivariate, and multivariate levels. Analysis was performed in STATA version 15.

Descriptive statistics of the background characteristics of the respondents were presented at univariate level. At bivariate level, Pearson's chi-squared ($\chi$2) tests were used to examine the significant differences between the nutritional outcomes of children and the explanatory variables. Multiple logistic regression models were fitted to determine the relationship between the three components of nutritional status and explanatory variables. Results are presented in the form of odds ratios with 95% confidence intervals of the variables.

## Ethics statement

This paper is based on data in the public domain. We sought permission to use the 2016 UDHS datasets from the DHS Program website (https://www.dhsprogram.com/data/dataset/Uganda_Standard-DHS_2016.cfm?flag=0). The ICF Institutional Review Board (IRB) reviewed and approved the 2016 UDHS. The ICF IRB complied with the United States Department of Health and Human Services regulations for the protection of human research subjects (45 CFR 46). Informed consent was obtained from participants and their participation was on voluntary basis. For purposes of maintenance of anonymity, participants' identifiers were not included in the dataset. Further details regarding the conduct of the study may be found in the 2016 UDHS report [10].

## Results

### Descriptive characteristics

Table 1 presents the descriptive results of this study. Results show that 28%, 4% and 11% of the children of the working women were stunted, wasted and underweight respectively. Slightly over 4 in every 5 households (81%) were rural households. Regions were proportionately represented in the sample. About three in ten of the working women (35%) and their children belonged to households in the rich wealth quintiles while four in ten of these women (45%) belonged to households in the poor quintiles. Only 17% of the households had improved toilet facilities. The biggest proportion of the households (76%) had access to improved sources of drinking water. About 26% of the households were female headed. The majority of households had less than 3 children U5 (77%), were below 35 years (76%), had primary or no formal education (74%) and were currently in union at the time of the survey (86%). Concerning occupation, over half of the women (56%) were engaged in agriculture work. Only 9% were in formal employment (professional and clerical jobs). Over half of the women (69%) were self-employed.

The majority of women (64%) did not participate in decision making concerning their health. Two in five women (42%) reported that distance to the health facility was a big problem in hindering their access to medical help. With respect to children characteristics, results show that 51% of the children were male. The majority were 24–59 months old (59%); and weighed 2.5 or more kilograms at birth (60%). About 61% of the children had received Vitamin A supplements and 55% had taken drugs for intestinal parasites during the six months preceding the survey. Over three in five children (65%) had fever, diarrhoea or cough in the 2 weeks before the survey.

Table 1 further shows the results of the cross tabulations between the child health outcomes (stunting, wasting and underweight) and selected explanatory factors.

**Stunting.** Stunting was significantly associated with type of residence, region, wealth index, type of toilet facility, source of drinking water, maternal age, education attainment, occupation, type of employer, child's age, birth weight and whether the child was dewormed. Stunting was higher among children in rural areas (29%) and the western region of the country (33%). Further, stunting was highest in households that had unimproved toilet facilities (31%) and unimproved sources of drinking water (32%). Stunting was also higher among children whose mothers were aged 15–24 years (30%), had no education (37%), were engaged in manual work (32%), and in children with less than 2.5 kilograms at birth (41%) and who were given drugs for intestinal parasites (30%). Sex of the household head, number of children in the household, marital status, participation in decision making concerning woman's own health, whether distance is a problem in accessing health services, sex of child, vitamin A supplementation and sickness were not significantly associated with stunting in children (Table 1).

**Wasting.** As shown in Table 1, wasting was significantly associated with region, wealth index of the household, sex of the child, age of the child and deworming. Higher levels of wasting were observed among children residing in households in the Northern region (8%), in the poorest wealth quintile (6%), among male children (5%), those age 0–23 months (7%) and those that had not taken drugs for deworming (5%). Residence, type of toilet facility, source of drinking water, sex of the household head, number of children in the household, maternal age, level of education, occupation, employer, involvement in decision making, distance as a problem in accessing health services, sex of child, birth weight, vitamin A supplementation and sickness were not significantly associated with wasting in children.

**Table 1. Distribution of children by their household, mothers' and demographic characteristics and their nutritional status.**

| VARIABLES/ CATEGORIES | % of total | Number of Children | % stunted | p-value | % wasted | p-value | % under-weight | p-value |
|---|---|---|---|---|---|---|---|---|
| Residence | | | | 0.039 | | 0.240 | | 0.016 |
| Urban | 19.3 | 681 | 24.1 | | 2.9 | | 7.3 | |
| Rural | 80.7 | 2850 | 29.1 | | 4.1 | | 11.3 | |
| Region | | | | 0.002 | | 0.000 | | 0.000 |
| Central | 24.2 | 853 | 24.5 | | 2.1 | | 7.2 | |
| East | 28.1 | 993 | 25.1 | | 3.5 | | 9.3 | |
| North | 22.6 | 798 | 30.3 | | 7.5 | | 14.6 | |
| West | 25.1 | 887 | 33.2 | | 2.8 | | 11.7 | |
| Wealth Index | | | | 0.000 | | 0.006 | | 0.000 |
| Poorest | 24.0 | 848 | 32.4 | | 5.9 | | 15.7 | |
| Poorer | 21.1 | 744 | 31.6 | | 4.6 | | 11.7 | |
| Middle | 19.7 | 697 | 31.3 | | 3.3 | | 10.6 | |
| Richer | 18.2 | 641 | 26.6 | | 2.2 | | 8.3 | |
| Richest | 17.0 | 601 | 15.8 | | 2.5 | | 3.8 | |
| Type of toilet facility | | | | 0.000 | | 0.144 | | 0.001 |
| Improved | 17.1 | 602 | 21.7 | | 2.8 | | 6.6 | |
| Shared facility | 16.2 | 570 | 22.4 | | 2.7 | | 7.4 | |
| unimproved toilet | 66.7 | 2358 | 31.2 | | 4.4 | | 12.2 | |
| Source of drinking water | | | | 0.026 | | 0.489 | | 0.723 |
| Improved | 76.4 | 2697 | 27.0 | | 4.0 | | 10.4 | |
| unimproved | 23.6 | 834 | 31.9 | | 3.4 | | 10.9 | |
| Sex of the household head | | | | 0.512 | | 0.053 | | 0.109 |
| Male | 74.3 | 2624 | 27.8 | | 3.5 | | 10.0 | |
| Female | 25.7 | 907 | 29.2 | | 5.0 | | 12.0 | |
| Children in Household | | | | 0.098 | | 0.375 | | 0.028 |
| 1 child | 29.4 | 1037 | 25.0 | | 3.1 | | 8.1 | |
| 2 Children | 48.1 | 1700 | 29.4 | | 4.0 | | 11.1 | |
| 3+ Children | 22.5 | 794 | 29.6 | | 4.5 | | 12.2 | |
| Mother's age | | | | 0.040 | | 0.991 | | 0.969 |
| 15–24 | 29.5 | 1041 | 30.0 | | 3.9 | | 10.7 | |
| 25–34 | 46.6 | 1645 | 29.1 | | 3.9 | | 10.4 | |
| 35–49 | 23.9 | 844 | 24.1 | | 3.8 | | 10.5 | |
| Mother's level of Education | | | | 0.000 | | 0.361 | | 0.000 |
| None | 10.9 | 386 | 36.2 | | 4.6 | | 16.7 | |
| Primary | 63.3 | 2237 | 29.8 | | 4.1 | | 11.3 | |
| Secondary+ | 25.8 | 909 | 20.8 | | 3.0 | | 5.9 | |
| Marital status | | | | 0.713 | | 0.499 | | 0.038 |
| Never in union | 3.2 | 112 | 26.8 | | 4.2 | | 11.9 | |
| Currently in union | 86.3 | 3048 | 28.0 | | 3.7 | | 9.9 | |
| Formerly in union | 10.5 | 370 | 30.2 | | 5.2 | | 14.7 | |
| Maternal Occupation | | | | 0.000 | | 0.285 | | 0.001 |
| Professional/Formal | 8.9 | 315 | 15.2 | | 3.3 | | 4.7 | |
| Sales and services | 14.9 | 526 | 21.9 | | 3.6 | | 7.3 | |
| Agriculture | 55.6 | 1958 | 30.9 | | 3.5 | | 11.6 | |
| Manual Work | 19.6 | 691 | 31.5 | | 5.5 | | 12.7 | |
| Domestic work | 1.0 | 34 | 24.3 | | 5.1 | | 7.0 | |

*(Continued)*

**Table 1.** (Continued)

| VARIABLES/ CATEGORIES | % of total | Number of Children | % stunted | p-value | % wasted | p-value | % under-weight | p-value |
|---|---|---|---|---|---|---|---|---|
| Mother's Employer | | | | 0.004 | | 0.214 | | 0.641 |
| Family member | 17.5 | 617 | 29.3 | | 4.3 | | 11.6 | |
| Nonfamily member | 13.5 | 477 | 21.0 | | 5.2 | | 11.1 | |
| Self employed | 69.0 | 2437 | 29.3 | | 3.5 | | 10.1 | |
| Mother involved in decision making on health | | | | 0.684 | | 0.158 | | 0.045 |
| No | 64.3 | 2271 | 27.6 | | 4.0 | | 9.7 | |
| Yes | 22.0 | 777 | 29.0 | | 2.7 | | 10.6 | |
| Not in Union | 13.7 | 483 | 29.4 | | 4.9 | | 14.0 | |
| Distance a problem to access to health services | | | | 0.452 | | 0.391 | | 0.335 |
| No | 58.0 | 2048 | 27.6 | | 3.6 | | 10.0 | |
| Yes | 42.0 | 1483 | 29.0 | | 4.2 | | 11.2 | |
| Sex of child | | | | 0.181 | | 0.020 | | 0.199 |
| Male | 50.7 | 1791 | 29.2 | | 4.7 | | 11.2 | |
| Female | 49.3 | 1740 | 27.0 | | 3.0 | | 9.8 | |
| Age of the child | | | | 0.044 | | 0.000 | | 0.042 |
| 0–23 months | 41.4 | 1462 | 26.2 | | 7.0 | | 11.8 | |
| 24–59 months | 58.6 | 2068 | 29.2 | | 1.7 | | 9.5 | |
| Child's birth weight | | | | 0.000 | | 0.066 | | 0.000 |
| < 2.5 Kgs | 6.1 | 215 | 41.3 | | 7.3 | | 21.4 | |
| 2.5 Kgs or more | 59.7 | 2108 | 25.4 | | 3.5 | | 8.1 | |
| Don't know/Not weighed at birth | 34.2 | 1209 | 30.5 | | 3.9 | | 12.8 | |
| Vitamin a | | | | 0.075 | | 0.253 | | 0.460 |
| No | 39.2 | 1378 | 26.2 | | 4.4 | | 9.9 | |
| Yes | 60.8 | 2140 | 29.2 | | 3.6 | | 10.8 | |
| Deworming | | | | 0.035 | | 0.030 | | 0.018 |
| No | 45.3 | 1601 | 26.2 | | 4.7 | | 12.0 | |
| Yes | 54.7 | 1930 | 29.8 | | 3.2 | | 9.2 | |
| Sickness in last two weeks | | | | 0.951 | | 0.549 | | 0.752 |
| No | 35.3 | 1246 | 28.1 | | 4.2 | | 10.2 | |
| Yes | 64.7 | 2284 | 28.2 | | 3.7 | | 10.6 | |
| Total/Percentage | 100.0% | 3531 | 28.2% | | 3.9% | | 10.5% | |

**Underweight.** Results in Table 1 show that being underweight was significantly associated with residence, region, household wealth, type of toilet facility, number of children in the household, maternal educational attainment, marital status, occupation, involvement in decision making, child's age, birth weight and deworming. Higher levels of being underweight were evident in rural areas (11%), the northern region (14%), the poorest wealth category (16%), children from households with unimproved toilet facilities (12%) and where mothers had three or more U5 children (12%). Being underweight was also higher in children whose mothers had no education (17%), were formerly in union (15%), engaged in manual work (13%) and were not involved in decision making concerning their own health (11%). Additionally, children aged 0–23 (12%), whose birth weight was less than 2.5 kilograms (21%) and those who had not received drugs for deworming (12%) had higher levels of being underweight. Source of drinking water, sex of the household head, maternal age, type of employer, whether distance was a problem in accessing health services, sex of the child, vitamin A supplementation and sickness were not significantly associated with being underweight in children.

## Association between maternal occupation, other characteristics and child nutrition outcomes

Table 2 shows the results of the multivariate logistic regression that examines the relationship between child health outcomes (stunting, wasting and underweight) and maternal occupation.

Results in Table 2 addressing stunting show that children born to older mothers age35-49 had lower odds of stunting compared with children born to younger mothers age 15–24 years (OR 0.69, 95% CI 0.56–0.86). Notably, odds of stunting were lower among children whose mothers had primary, secondary, or higher levels of education compared with children whose mothers had no formal education (OR 0.78, 95% CI 0.62–0.97 and OR 0.64, 95% CI 0.47–0.88 respectively).

Children who weighed 2.5 kilogrammes or more at birth had lower odds of stunting compared with children who weighed less than 2.5 kilograms at birth (OR 0.59, 95% CI 0.45–0.78). Maternal employment was a significant determinant of stunting. Children whose mothers engaged in agriculture and manual work had higher odds of stunting compared with children whose mothers engaged in formal employment -professional/clerical jobs (OR 2.00, 95% CI 1.26–3.19 and OR 2.00, 95% CI 1.27–3.14). Children who were given drugs for deworming had higher odds of stunting than those who were not given (OR 1.18, 95% CI 1.00–1.39).

Analysis of weight for height i.e. wasting, showed that children in the northern region had higher odds of wasting compared with children in Central region (OR 4.04, 95% CI 2.15–8.46). Lower odds of wasting were observed among females compared with male children (OR 0.63, 95% CI 0.44–0.92); children above 2 years—24–59 months—compared with children below 2 years (OR 0.21, 95% CI 0.13–0.32); and children who weighed 2.5 kilograms or more at birth compared with those who weighed less than 2.5 kilograms at birth (OR 0.48, 95% CI 0.25–0.92). With respect to type of employer, children whose mothers were employed by a nonfamily member had higher odds of wasting compared with those whose mothers were employed by a family member (OR 2.30, 95% CI 1.18–4.46).

Results in Table 2 reveal that children in the Northern region had higher odds of being underweight compared with children in the Central region (OR 1.70, 95% CI 1.05–2.74). Lower odds of being underweight were evident among children whose mothers had secondary or higher levels of education compared with children whose mothers had no education (OR 0.52, 95% CI 0.31–0.84); children who weighed 2.5 kilograms or more at birth compared with children who weighed less than 2.5 kilograms at birth (OR 0.39, 95% CI 0.26–0.58). Children whose mothers were employed by nonfamily members had increased odds for being underweight compared with those whose mothers were employed by a family member (OR 1.77, 95% CI 1.02–3.07).

## Discussion

This study examined the determinants of nutrition status of U5 children. Significant determinants of child nutrition status were region, maternal age, education level, occupation/employment, employer, sex of child, age of child and birth weight.

Our findings are in agreements with related to studies in sub Saharan Africa that also found that children whose mothers were engaged in professional, technical and managerial work had better nutrition outcomes compared with other occupation categories [35].Women in formal have regular access to earnings that not only enhance their financial status and autonomy but most importantly contribute to child feeding [36]. Findings of the current study show that engaging in agricultural and manual work significantly increased the odds of stunting among children. In Uganda, women engaged in this sector often belong to poorer households which may have challenges in feeding for both the mothers and their children [37]. This has been

**Table 2. Adjusted odds ratios for malnutrition among children age 0–59 months in Uganda.**

| | STUNTING | | WASTING | | UNDERWEIGHT | |
|---|---|---|---|---|---|---|
| | Odds Ratio | (95%CI) | Odds Ratio | (95%CI) | Odds Ratio | (95%CI) |
| Residence (rc: Urban) | | | | | | |
| • Rural | 0.95 | 0.74–1.22 | 1.17 | 0.61–2.26 | 1.07 | 0.72–1.62 |
| Region (rc: Central) | | | | | | |
| • East | 0.90 | 0.69–1.18 | 1.73 | 0.83–3.46 | 1.06 | 0.67–1.67 |
| • North | 1.03 | 0.79–1.34 | 4.04*** | 2.15–8.46 | 1.70* | 1.05–2.74 |
| • West | 1.22 | 0.94–1.60 | 1.25 | 0.61–2.57 | 1.37 | 0.88–2.13 |
| Type of toilet facility (rc: Improved) | | | | | | |
| • Shared toilet | 1.02 | 0.72–1.46 | 0.85 | 0.36–2.00 | 1.17 | 0.69–1.98 |
| • Unimproved toilet | 1.16 | 0.88–1.53 | 1.14 | 0.60–2.17 | 1.25 | 0.79–1.98 |
| Children in Household (rc: I child) | | | | | | |
| • 2 Children | 1.15 | 0.94–1.39 | 1.19 | 0.74–1.91 | 1.30 | 0.98–1.74 |
| • 3+ Children | 1.14 | 0.91–1.42 | 1.18 | 0.69–2.01 | 1.32 | 0.95–1.85 |
| Age of mother (rc: 15–24) | | | | | | |
| • 24–34 | 0.96 | 0.81–1.14 | 1.36 | 0.83–2.21 | 1.05 | 0.78–1.41 |
| • 35–49 | 0.69*** | 0.56–0.86 | 1.38 | 0.79–2.42 | 0.96 | 0.68–1.36 |
| Mother's level of Education (rc: None) | | | | | | |
| • Primary | 0.78* | 0.62–0.97 | 1.10 | 0.56–2.16 | 0.73 | 0.52–1.02 |
| • Secondary+ | 0.64** | 0.47–0.88 | 0.97 | 0.40–2.34 | 0.52** | 0.31–0.84 |
| Marital status (rc: Never in union) | | | | | | |
| • Currently in union | 0.99 | 0.66–1.50 | 1.00 | 0.32–3.15 | 0.80 | 0.43–1.49 |
| • Formerly in union | 1.09 | 0.68–1.77 | 1.50 | 0.42–5.34 | 1.15 | 0.58–2.29 |
| Maternal Occupation (rc: Professional/Formal) | | | | | | |
| • Sales and services | 1.39 | 0.86–2.26 | 1.10 | 0.44–2.73 | 1.30 | 0.57–2.95 |
| • Agriculture | 2.00** | 1.26–3.19 | 0.76 | 0.32–1.78 | 1.83 | 0.81–4.11 |
| • Manual work | 2.00** | 1.27–3.14 | 1.14 | 0.48–2.73 | 1.79 | 0.82–3.87 |
| • Domestic work | 1.76 | 0.56–5.51 | 1.02 | 0.14–7.26 | 0.85 | 0.21–3.87 |
| Mother's employer (rc: Family member) | | | | | | |
| • Nonfamily member | 0.92 | 0.68–1.26 | 2.05* | 1.04–4.05 | 1.77* | 1.02–3.07 |
| • Self employed | 1.02 | 0.85–1.24 | 1.21 | 0.75–1.97 | 1.09 | 0.76–1.56 |
| Whether distance is a problem in accessing health services (rc: No) | | | | | | |
| • Yes | 0.98 | 0.83–1.14 | 1.05 | 0.71–1.56 | 0.96 | 0.75–1.22 |
| Sex of child (rc: Male) | | | | | | |
| • Female | 0.90 | 0.79–1.03 | 0.63* | 0.44–0.92 | 0.84 | 0.67–1.05 |
| Age of child in months (rc: 0–23 months) | | | | | | |
| • 24–59 months | 1.16 | 1.00–1.36 | 0.21*** | 0.13–0.32 | 0.81 | 0.64–1.04 |
| Child's birth weight (rc: < 2.5 Kgs) | | | | | | |
| • 2.5 Kgs or more | 0.59*** | 0.45–0.78 | 0.48* | 0.25–0.92 | 0.39*** | 0.26–0.58 |
| • Don't know/Not weighed at birth | 0.64** | 0.48–0.86 | 0.59 | 0.28–1.24 | 0.54** | 0.35–0.84 |
| Vitamin A (rc: No) | | | | | | |
| • Yes | 1.08 | 0.92–1.26 | 0.80 | 0.52–1.22 | 1.22 | 0.93–1.61 |
| Deworming (rc: No) | | | | | | |
| • Yes | 1.18* | 1.00–1.39 | 1.10 | 0.69–1.74 | 0.82 | 0.63–1.06 |
| Sickness in last two weeks (rc: No) | | | | | | |
| • Yes | 1.05 | 0.90–1.23 | 0.72 | 0.48–1.07 | 1.02 | 0.78–1.33 |

(*Continued*)

**Table 2.** (Continued)

| | STUNTING | | WASTING | | UNDERWEIGHT | |
|---|---|---|---|---|---|---|
| | Odds Ratio | (95%CI) | Odds Ratio | (95%CI) | Odds Ratio | (95%CI) |
| Total observations | 3584 | | | | | |

Level of significance:

\*\*\* p<0.001

\*\* p<0.01

\* p<0.05

CI: Confidence Interval

rc: Reference Category

attributed to commercialization of staple foods leading to household food insecurity. Additionally, the demands of agricultural work often leave limited time for child feeding [38]. It is likely that such women lack adequate nutrition knowledge that could be beneficial to child feeding [39].

Concerning the type of employer, employment by a family member had a mitigating effect on wasting and being underweight. Employment by a family member often entails empathy on the part of the employer and therefore flexibility to accommodate child care demands including child feeding [40, 41] which many not be feasible for mothers employed by non-family members [32, 42]. This is likely to increase breastfeeding for such children and consequently improve child growth and development [43, 44]

Younger children (0–24 months) are more vulnerable to wasting. It is possible that this results from infections in contaminated feed and unsterilized feeding equipment [45, 46]. Exposure to contaminated food increases diarrhoeal diseases which results into poor nutritional indicators. Another possible explanation for the high levels of wasting is malaria in children, which is one of the leading causes of morbidity and mortality in Uganda [47].

Results showed that children in the Northern region had higher odds of wasting and being underweight compared with children in the Central region of Uganda. The Northern region of Uganda consists of some of the most socially and economically disadvantaged sub regions of Uganda [48]. This region is affected by food insecurity and the negative effects of the civil war. Though this study and Ajao, Ojofeitimi [49] found no association between the number of U5 children and nutrition outcomes, the number of children in the household may influence the quality of care including the amount of food that is dedicated to each child. In cases where the household has a large number of children and is economically disadvantaged, children may be deprived of food and this may lead to poor nutritional outcomes.

Our results showed that children of more educated women had better nutritional outcomes than those of uneducated women. This may partially be explained by the increase in health knowledge, autonomy and empowerment among educated women [50, 51]. Women's autonomy in the household directly affects their involvement in the nutrition of their household members. This has the potential of improving nutrition outcomes of children especially those that are given complementary foods [33, 52]. Educated women have access to nutritional information, and this influences diet, feeding patterns and health seeking behaviour [53]. Such information leads to healthier food choices such as more intake of fruits vegetables and legumes. Education also increases use of health services that are important in not only improving the lives of the children but also their mothers [54]. Education further increases the age at first birth especially among women who attain secondary education [55] and this is likely to lead to better child health outcomes.

Children born to older women had lower odds of stunting than children of younger women. A possible explanation for the better nutritional outcomes among children of older women, is that older mothers may have more experience in child care than the younger mothers. These findings are similar to findings by Nigatu, Assefa Woreta [23] and Fentahun, Wubshet [56].

Sex of the child was significantly associated with wasting. Female children showed reduced odds of wasting than male children. This finding is similar to findings elsewhere that show better nutritional outcomes for males compared with females [57, 58]. This could be explained by the fact that due to the genetic makeup of female children; they can easily survive in limited food supply settings than male children.

Similar to findings elsewhere [59], child's birth weight was significantly associated with all the three nutrition outcomes. Bigger children had lower odds of stunting, wasting and being underweight than smaller children. Malnourished mothers are more likely to deliver underweight children and conditions that contribute to such situations are likely to persist after delivery.

Whereas studies elsewhere found that deworming was associated with reduced stunting among children [60], this study found that deworming increased the odds of stunting. It is possible that deworming was prompted by the childrens poor nutrition/health conditions. The drugs may have been administeres as part of clinical management of childhood illnesses.

This study had some limitations that need to be acknowledged. The UDHS data are cross-sectional survey, and are therefore limited in their ability to assess casual relationships between variables. Most of the data were generated from self-reported information provided by survey respondents; it is therefore subject to respondent bias. The study did not include information about when the respondent started to work; it was therefore not possible to ascertain the duration in employment yet this may influence child health. However, the study offers vital infromation on maternal employment and child nutrition, that could be a basis for programmatic response.

## Conclusion

In Uganda, maternal employment is a significant determinant of child nutritional status. Other significant determinants are region, maternal age and education, and child's age and weight at birth. The most vulnerable children are those whose mothers belong to the low socioeconomic status, are young, reside in places that are food insecure and have low levels of education.

Interventions to increase maternal labour force participation should advocate for flexible work schedules—so that mothers can either leave work earlier or report later to involve in childcare. This is likely to support exclusive and continued breastfeeding, and consequently improve retention of women in employment. Interventions to improve the socioeconomic situation and nutrition of the population should target Northern Uganda in order to neutralize the palpable effects of the war. The study underscores the importance of education in improving the lives of individuals and their children. Efforts towards ANC and skilled delivery should place more emphasis on nutrition education in the antenatal and postnatal periods, to sensitize mothers about the benefits and dietary content of proper child and maternal feeding, and mosquito net use to prevent malaria. These interventions should have special focus on young mothers. Further research is needed to explain the reasons for the poor nutritional indicators among these categories in Uganda.

## Acknowledgments

The authors are grateful to Makerere University (Department of Population Studies and Department of Social Work) for creating an enabling environment that made it possible to

carry on this research study. We thank the DHS program for the permission to use the UDHS data.

## Author Contributions

**Conceptualization:** Olivia Nankinga, Betty Kwagala.

**Formal analysis:** Olivia Nankinga, Betty Kwagala.

**Methodology:** Olivia Nankinga, Betty Kwagala, Eddy J. Walakira.

**Validation:** Olivia Nankinga, Eddy J. Walakira.

**Writing – original draft:** Olivia Nankinga.

**Writing – review & editing:** Olivia Nankinga, Betty Kwagala, Eddy J. Walakira.

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
