## [Decision Letter · Decision Letter 0]

7 Oct 2019

PONE-D-19-22513

Maternal Employment and Child Nutritional Status in Uganda

PLOS ONE

Dear Mrs Nankinga,

Thank you for submitting your manuscript to PLOS ONE. After careful consideration, we feel that it has merit but does not fully meet PLOS ONE’s publication criteria as it currently stands. Therefore, we invite you to submit a revised version of the manuscript that addresses the points raised during the review process.

We would appreciate receiving your revised manuscript by Nov 21 2019 11:59PM. To enhance the reproducibility of your results, we recommend that if applicable you deposit your laboratory protocols in protocols.io, where a protocol can be assigned its own identifier (DOI) such that it can be cited independently in the future. For instructions see: http://journals.plos.org/plosone/s/submission-guidelines#loc-laboratory-protocols

We look forward to receiving your revised manuscript.

Kind regards,

Kannan Navaneetham

Academic Editor

PLOS ONE

Journal Requirements:

Additional Editor Comments (if provided):

Reviewers' comments:

Reviewer's Responses to Questions

**Comments to the Author**

1. Is the manuscript technically sound, and do the data support the conclusions?

Reviewer #1: Partly

Reviewer #2: Partly

2. Has the statistical analysis been performed appropriately and rigorously? 

Reviewer #1: Yes

Reviewer #2: Yes

3. Have the authors made all data underlying the findings in their manuscript fully available?

Reviewer #1: Yes

Reviewer #2: Yes

4. Is the manuscript presented in an intelligible fashion and written in standard English?

Reviewer #1: No

Reviewer #2: Yes

5. Review Comments to the Author

Reviewer #1: Please see attached comments.

Reviewer #2: In this paper, the authors try to relate the maternal employment with child nutritional outcomes using the cross-sectional DHS data conducted in Uganda during 2016. The authors presented univariate, bivariate and multivariate analyses to fulfil their objective. The analytical methods for the paper are rightly executed. My observations on the manuscript are following.

1. In the introduction, the authors highlighted the role of child malnutrition in child morbidity and mortality. Also, it describes the associated factors of child health outcomes. The focus on the review of the mother’s employment and child nutritional outcomes is limited. Hence, the authors should add more details about the previous literature.

2. In the data and method section, more clarification is needed on the sample selection for employment. A coordinated expansion of sample selection with the following statement is required.

“Employed women included women who worked in the 7 days preceding the survey and those who did not work in the 7 days but are regularly employed and were absent from work for leave, illness, vacation or any related reason.”

3. In this line, can the association between employment status in the last seven days and the child nutritional status be justified? The under-five children have an exposure of seven days to the mother’s employment status, while the nutritional outcomes of children are the exposure of 0 to 5 years.

4. A detailed categorisation plan of maternal occupation/employment variable is required with sample size (Page 6, line no. 139).

5. The types of works may be varied by employers. A description of the type of works by different employers will educate more to the readers.

6. The association of child undernutrition with the variable of interaction between occupational status and employer could give an interesting insight to the readers.

7. In the title of table 2, the sample is the children aged 6-59 months. In the variable, the age of children in the same table, the sample is the children aged 0-59 months. Please explains the inconsistency.

8. In the page no. 14 and line no. 295, the reporting of OR is confusing/not clear.

9. In the page no. 15 and line no. 333, elaborate the linkage of commercialisation and selling of products. Further, in this point, the manual labourers could have different mechanisms of the linkage with child undernutrition as compared to the agricultural labourers. So, the authors should separate the category of agricultural labour and manual labour in the variable.

10. In the page no. 16 and line no. 346-347, The statement needs elaboration and citation.

11. In the page no. 16 and line 356-58, “This can be reinforced by putting in place preschools by the employers for children that may be older than 24 months in order to support maternal involvement in child care”. A significant share of the children belongs to the poor wealth quintile of the study region. Also, the manual and agricultural workers may not afford the cost of pre-schooling for their children. So, it is hard to draw a conclusion that the employed mothers would have sent their children to the pre-school.

12. Moreover, in writing the results and discussion, the objective should get more attentions rather than the other independent variables. Also, the manuscript needs an English academic editorial exercise.

6. PLOS authors have the option to publish the peer review history of their article (what does this mean?). If published, this will include your full peer review and any attached files.

Reviewer #1: No

Reviewer #2: Yes: Md Juel Rana

---

## [Author Response · Author response to Decision Letter 0]

19 Nov 2019

REVIEWER 1 COMMENTS AND AUTHORS’ RESPONSES

1 One of my major concerns about the paper was that certain statements made in the discussion section do not match the results presented in Table 2. For e.g.:

a)in Pg 15 lines 327-329:

‘Related to studies elsewhere, this study showed that children whose mothers were engaged in professional, technical and managerial work had lower odds of stunting and underweight compared with other occupation categories.’

However, the results from table 2 show that children whose mothers were engaged in professional, technical and managerial work had significantly lower odds of stunting compared with agricultural and manual workers (not all other occupations), also results for underweight were not significant.

b) Line 392-393 (Pg 18) says ‘Female children showed reduced odds of stunting than male children’ 

This should be wasting and not stunting.

It would be great if the authors double check the results reported in the discussion section with the Table 2. 

We have revised the discussion of the paper to match with the results.

We have also double checked the results and discussion as advised by the reviewer

2 Could multiple children in the analyses have the same mother? Is this taken into account in the regression?

 We wish to acknowledge the fact raised by the reviewer, however according to Rutstein*, analysts are advised that instead of using data on only one index child, considering all data on children under age 5 would reduce possible bias in the results.

*: Rutstein, Shea O. 2014. Potential Bias and Selectivity in Analyses of Children Born in the Past Five Years Using DHS Data. DHS Methodological Reports No. 14. Rockville, Maryland, USA: ICF International.

3 There should be a clear link between the second last paragraph and the last paragraph in the introduction section (pg 4, line 90-99) and Pg 5 line 101-102. Perhaps adding a paragraph or two on: more studies on maternal employment and child health, identifying gaps in the literature, explaining their contribution clearly, and highlighting that they are one of the first to examine the relationship between maternal employment, and child health outcomes in Uganda. Also, lines 101-106 (Pg 5) do not lead up to the research question, and could be omitted. 

We have revised and linked the paragraphs as has been advised by the reviewer. 

 Minor comments 

4 Pg 3 lines 53-60, please give reference of the data source. 

The manuscript includes the references for the estimates.

These are UNICEF (United Nations Children's Fund) and WHO (World Health Organization) and are marked as references 2-4 as seen in the paragraph. These have been reorganized to improve readability

5 Pg 5: line 117: What is the full form of UGKR? 

This is the children data file.

We have included this information under the data source section as suggested.

6 Minor clarification: Pg 7 line 156: refers to those recoded to 2? 

Yes, this is in reference to category 2,

We have revised it to read better.

7 In the discussion section the authors could consider linking study conclusion to other studies in Sub Saharan Africa that addressed similar questions. Some studies that the authors might find useful are: 

We have read the suggested studies and linked them to our study.

8 8. Pg 18 line 386-387: reframe the sentence to say that ‘children born to older women have better health outcomes’ instead of the existing sentence that says ‘older women have better health outcomes than younger women’. 

The sentence has been changed.

 REVIEWER 2 COMMENTS AND AUTHORS’ RESPONSES

1 In the introduction, the authors highlighted the role of child malnutrition in child morbidity and mortality. Also, it describes the associated factors of child health outcomes. The focus on the review of the mother’s employment and child nutritional outcomes is limited. Hence, the authors should add more details about the previous literature.

More details on mother’s employment and child nutritional status have been added as advised by the reviewer

2 In the data and method section, more clarification is needed on the sample selection for employment. A coordinated expansion of sample selection with the following statement is required.

“Employed women included women who worked in the 7 days preceding the survey and those who did not work in the 7 days but are regularly employed and were absent from work for leave, illness, vacation or any related reason.”

The sample selection procedure has been expanded.

3 In this line, can the association between employment status in the last seven days and the child nutritional status be justified? The under-five children have an exposure of seven days to the mother’s employment status, while the nutritional outcomes of children are the exposure of 0 to 5 years.

 From the data, it is not possible to know when the woman began work. This is a weakness of the question used. But the question captures all people who are employed. The question captured occasional, seasonal and those employed throughout the year. But excluding the seasonal and occasional does not change the results significantly. We want to believe that the question is aimed at capturing everyone in work (in the Uganda context) but acknowledge that this may differ on how employment is captured elsewhere 

In addition, we ran a model for with only mothers who were employed throughout the year. However, the results do not differ significantly.

We attach, the results from the analysis conducted with only those employed throughout the year.

4 A detailed categorisation plan of maternal occupation/employment variable is required with sample size (Page 6, line no. 139).

 We have done this as advised. 

5 The types of works may be varied by employers. A description of the type of works by different employers will educate more to the readers.

 This has been done.

6 The association of child undernutrition with the variable of interaction between occupational status and employer could give an interesting insight to the readers.

 We had initially provided some information and more details have been added. 

7 In the title of table 2, the sample is the children aged 6-59 months. In the variable, the age of children in the same table, the sample is the children aged 0-59 months. Please explains the inconsistency.

 The sample is for children age 0-59 months. The inconsistency has been corrected.

8 In the page no. 14 and line no. 295, the reporting of OR is confusing/not clear.

 The reported odds are for mother’s education and stunting.

We stated both odds ratios and 95% confidence intervals for the results as shown in Table 2.

9 In the page no. 15 and line no. 333, elaborate the linkage of commercialisation and selling of products. Further, in this point, the manual labourers could have different mechanisms of the linkage with child undernutrition as compared to the agricultural labourers. So, the authors should separate the category of agricultural labour and manual labour in the variable.

 We have separated the category as advised by the reviewer

10 In the page no. 16 and line no. 346-347, The statement needs elaboration and citation.

 It has been elaborated as advised.

11 In the page no. 16 and line 356-58, “This can be reinforced by putting in place preschools by the employers for children that may be older than 24 months in order to support maternal involvement in child care”. A significant share of the children belongs to the poor wealth quintile of the study region. Also, the manual and agricultural workers may not afford the cost of pre-schooling for their children. So, it is hard to draw a conclusion that the employed mothers would have sent their children to the pre-school.

 This result has since changed. And the content has been edited.

12 Moreover, in writing the results and discussion, the objective should get more attentions rather than the other independent variables. Also, the manuscript needs an English academic editorial exercise. 

We have done this as advised by the reviewer

---

## [Decision Letter · Decision Letter 1]

5 Dec 2019

Maternal Employment and Child Nutritional Status in Uganda

PONE-D-19-22513R1

Dear Dr. Nankinga,

We are pleased to inform you that your manuscript has been judged scientifically suitable for publication and will be formally accepted for publication once it complies with all outstanding technical requirements.

With kind regards,

Kannan Navaneetham

Academic Editor

PLOS ONE

Additional Editor Comments (optional):

Reviewers' comments:

Reviewer's Responses to Questions

**Comments to the Author**

1. If the authors have adequately addressed your comments raised in a previous round of review and you feel that this manuscript is now acceptable for publication, you may indicate that here to bypass the “Comments to the Author” section, enter your conflict of interest statement in the “Confidential to Editor” section, and submit your "Accept" recommendation.

Reviewer #2: All comments have been addressed

2. Is the manuscript technically sound, and do the data support the conclusions?

Reviewer #2: Yes

3. Has the statistical analysis been performed appropriately and rigorously? 

Reviewer #2: Yes

4. Have the authors made all data underlying the findings in their manuscript fully available?

Reviewer #2: Yes

5. Is the manuscript presented in an intelligible fashion and written in standard English?

Reviewer #2: Yes

6. Review Comments to the Author

Reviewer #2: I was seeking an elaboration about manual worker and agricultural worker in the conclusion. How do the existing programmes can be modified or the new programme can be introduced to address the issues of child under-nutrition of these working mothers.

7. PLOS authors have the option to publish the peer review history of their article (what does this mean?). If published, this will include your full peer review and any attached files.

Reviewer #2: Yes: Md. Juel Rana

---

## [Editor Report · Acceptance letter]

10 Dec 2019

PONE-D-19-22513R1 

Maternal Employment and Child Nutritional Status in Uganda 

Dear Dr. Nankinga:

I am pleased to inform you that your manuscript has been deemed suitable for publication in PLOS ONE. Congratulations! Your manuscript is now with our production department. 

With kind regards,

on behalf of

Professor Kannan Navaneetham 

Academic Editor

PLOS ONE